# TOWARDS UNDERSTANDING HOW MOMENTUM IMPROVES GENERALIZATION IN DEEP LEARNING

## ABSTRACT

Stochastic gradient descent (SGD) with momentum is widely used for training modern deep learning architectures. While it is well understood that using momentum can lead to faster convergence rate in various settings, it has also been observed that momentum yields higher generalization. Prior work argue that momentum stabilizes the SGD noise during training and this leads to higher generalization. In this paper, we take the opposite view to this result and first empirically show that gradient descent with momentum (GD+M) significantly improves generalization comparing to gradient descent (GD) in many deep learning tasks. From this observation, we formally study how momentum improves generalization in deep learning. We devise a binary classification setting where a two-layer (over-parameterized) convolutional neural network trained with GD+M provably generalizes better than the same network trained with vanilla GD, when both algorithms start from the same random initialization. The key insight in our analysis is that momentum is beneficial in datasets where the examples share some features but differ in their margin. Contrary to the GD model that memorizes the small margin data, GD+M can still learn the features in these data thanks to its historical gradients. We also empirically verify this learning process of momentum in real-world settings.

## 1 INTRODUCTION

It is commonly accepted that adding momentum to an optimization algorithm is required to optimally train a large-scale deep network. Most of the modern architectures maintain during the training process a heavy momentum close to 1 (Krizhevsky et al., 2012; Simonyan & Zisserman, 2014; He et al., 2016; Zagoruyko & Komodakis, 2016). Indeed, it has been empirically observed that architectures trained with momentum outperform those which are trained without (Sutskever et al., 2013). Several papers have attempted to explain this phenomenon. From the optimization perspective, Defazio (2020) assert that momentum yields faster convergence of the training loss since, at the early stages, it cancels out the noise from the stochastic gradients. On the other hand, Leclerc & Madry (2020) empirically observes that momentum yields faster training convergence only when the learning rate is small. While these works shed light on how momentum acts on neural network training, they fail to capture the generalization improvement induced by momentum (Sutskever et al., 2013). Besides, the noise reduction property of momentum advocated by Defazio (2020) seems to even contradict the observation that, in deep learning, having a large noise in the training improves generalization (Li et al., 2019; HaoChen et al., 2020). To the best of our knowledge, there is no existing work which *theoretically explains* how momentum improves generalization in deep learning. Therefore, this paper aims to close this gap and addresses the following question:

*Is the higher generalization induced by momentum tied to the stochastic noise of the gradient? If not, what is the underlying mechanism of momentum improving generalization in deep learning?*

In this paper, we empirically verify that the generalization improvement induced by momentum *is not* tied to the stochasticity of the gradient. Indeed, as reported in Figure 1, momentum improves generalization more significantly for full batch GD than for SGD in CIFAR object recognition tasks. Motivated by this empirical observation and the fact that the stochastic noise influences generalization, we theoretically study how gradient descent with momentum (GD+M) can generalize better than vanilla gradient descent (GD). We therefore *only focus on the contribution of momentum of the true gradient on generalization*.

The question we address concerns algorithmic regularization which characterizes the generalization of an optimization algorithm when multiple global solutions exist in over-parameterized deep learning

| | CIFAR-10 | | CIFAR-100 | |
|---|---|---|---|---|
| | Test | Ratio | Test | Ratio |
| R18 | 75.83/84.68 | 1.11 | 43.32/51.99 | 1.20 |
| WR16 | 75.02/84.48 | 1.12 | 42.95/51.33 | 1.20 |

(a)

| | CIFAR-10 | | CIFAR-100 | |
|---|---|---|---|---|
| | Test | Ratio | Test | Ratio |
| R18 | 86.15/85.91 | 0.99 | 53.81/58.01 | 1.08 |
| WR16 | 84.83/87.85 | 1.04 | 55.09/60.83 | 1.10 |

(b)

Figure 1: Test accuracy obtained with Resnet-18 (R18) and WideResnet16 (WR16) on CIFAR-10 and CIFAR-100. The architectures are trained using GD/GD+M (a) and SGD/ SGD+M (b) for 300 epochs to ensure zero training error. (c)-(d) respectively display the training loss and test accuracy by R18 with GD/GD+M on CIFAR-10. To isolate the effect of momentum, we *turn off* data augmentation, dropout and batch normalization. GD and SGD respectively refer to stochastic gradient descent with batch sizes $50k$ (full batch) and $128$. We grid searched the best (scheduled) learning rate and weight decay for each individual algorithm separately. Results are averaged over 3 runs and we only report the mean (see Appendix for complete table).

model Soudry et al. (2018); Lyu & Li (2019); Ji & Telgarsky (2019); Chizat & Bach (2020); Gunasekar et al. (2018); Arora et al. (2019). This regularization arises in deep learning mainly due to the *non-convexity* of the objective function. Indeed, this latter can create multiple global minima scattered in the space that vastly differ in terms of generalization. Algorithmic regularization is induced by and depends on many factors such as learning rate and batch size (Goyal et al., 2017; Hoffer et al., 2017; Keskar et al., 2016; Smith et al., 2018), initialization Allen-Zhu & Li (2020), adaptive step-size (Kingma & Ba, 2014; Neyshabur et al., 2015; Wilson et al., 2017), batch normalization (Arora et al., 2018; Hoffer et al., 2019; Ioffe & Szegedy, 2015) and dropout (Srivastava et al., 2014; Wei et al., 2020). However, none of these works theoretically analyzes the regularization induced by momentum. We therefore start our investigation by raising the following question:

*Does momentum **unconditionally** improve generalization in deep learning?*

This question could be positively answered given the success of momentum for learning distinct architectures such as ResNets (He et al., 2016) or BERT (Devlin et al., 2018). However, we here empirically give a negative answer through the following synthetic example in deep learning. We consider a binary classification problem where data-points are generated from a standard normal distribution and labels are outputs of teacher networks. Starting from the same initialization, we train different over-parametrized student networks using GD and GD+M. Based on Table 1, whether the target function is simple (linear) or complex (neural network), momentum does not improve generalization even when using a non-linear neural network as learner. The same observation holds for SGD/SGD+M as shown in the Appendix. Therefore, momentum *does not* always lead to a higher generalization in deep learning. Instead, such benefit seems to heavily depend on both the *structure of the data* and the *learning problem*.

**On which data set does momentum help generalization?** In this paper, **in order to determine the underlying mechanism produced by momentum to improve generalization**, we design a binary classification problem with a simple data structure where training a two-layer (over-parametrized) convolutional network with momentum provably improves generalization in deep learning. It is built upon a data distribution that relies on the concepts of *feature* and *margin*. Informally, each example in this distribution is a 1D image having $P$ patches. One of the patches (the signal patch) contains a feature we want to learn and all the others are Gaussian random noise with small variance.

Mathematically, one can think of a feature as a vector $w^* \in \mathbb{R}^d$. We assume that our training examples are divided into *large margin* data where the signal is $\alpha w^*$ with $\alpha$ constant and *small margin* data where the signal is $\beta w^*$ with $\beta \ll 1$. Intuitively, the second type of data is inherently noisier as the margin is small and therefore, a classifier would struggle more to generalize on this type of data. We

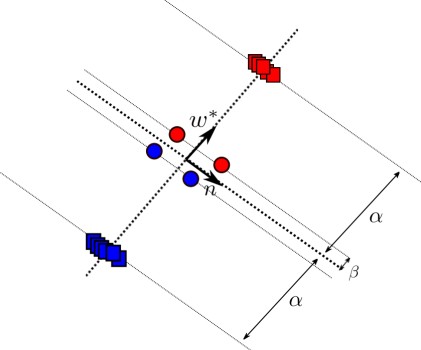

Figure 2: Dataset equation (D) 2D. Each data-point is $X_i = [c_i \cdot w^*, d_i \cdot n] \in \mathbb{R}^4$ for some $c_i, d_i \in \mathbb{R}$. We project these points in the 2D space $(\mathrm{span}(w^*), \mathrm{span}(n))$. The feature is $w^*$ and the noisy patch is in $\mathrm{span}(n)$. The large margin data (squares) have large component along $w^*$ and relatively small noise component and are thus roughly equal to $\alpha w^*$. The small margin data (circles) have relatively large noise component and thus, these data are well-spread on the span of $n$.

| Teacher / Student | Linear | 1-MLP | 2-MLP | 1-CNN | 2-CNN |
|---|---|---|---|---|---|
| 1-MLP | 93.48/93.25 | 92.32/92.18 | 84.3/83.68 | 94.18/94.12 | 76.04/76.12 |
| 2-MLP | 93.45/92.85 | 91.02/91.78 | 83.82/83.25 | 94.14/94.20 | 75.50/75.56 |
| 1-CNN | 92.21/92.34 | 92.31/92.33 | 83.39/83.44 | 94.39/94.39 | 79.44/78.32 |
| 2-CNN | 91.04/91.22 | 91.51/91.56 | 82.44/82.12 | 93.91/93.79 | 80.86/78.56 |

Table 1: Test accuracy obtained using GD/GD+M on a Gaussian synthetic dataset trained using neural network with ReLU activations. The training dataset consists in 500 data points in dimension 30 and test set in 5000 points. The student networks are trained for 1000 epochs to ensure zero training error. The results are averaged over 3 runs and we only report the mean (see Appendix for complete table).

underline that all the examples share the *same feature* but differ in the intensity of the signal. We consider a training dataset of size $N$ with the following split for $\hat{\mu} \ll 1$ :

$$(1 - \hat{\mu})N \text{ datapoints are with large margin,}$$
$$\hat{\mu}N \text{ datapoints are with small margin data.} \tag{D}$$

Figure 2 sketches equation (D) in a 2D setting. We emphasize that datasets having similar features and different margins are common in the real-world. Examples include object-recognition datasets such as CIFAR (Krizhevsky et al., 2009) or Imagenet (Deng et al., 2009) (for example, the "wheel feature" of a car can be strong or weak depending on the orientation of the car). More specifically, we believe that the dataset (D) can be viewed as a simplified model of these object-recognition datasets. In this context, the following informal theorems characterize the generalization of the GD and GD+M models. They dramatically simplify Theorem 3.1 and Theorem 3.2 but highlight the intuitions behind our results.

**Theorem 1.1** (Informal, GD+M). *There exists a dataset of the form (D) with size $N$ such that a two-layer (over-parameterized) convolutional network trained with GD+M:*

    *1. initially only learns large margin data from the $(1 - \hat{\mu})N$ examples.*

    *2. has large historical gradients that contain the feature $w^*$ present in small margin data.*

    *3. keeps learning the feature in the small margin data using its momentum historical gradients.*

*The model thus reaches zero training error and perfectly classify large and small margin data at test.*

**Theorem 1.2** (Informal, GD). *There exists a dataset of the form (D) with size $N$ such that a two-layer (over-parameterized) convolutional network trained with GD:*

    *1. initially only learns large margin data from the from the $(1 - \hat{\mu})N$ examples.*

    *2. has small gradient after learning these data.*

    *3. memorizes the remaining small margin data from the $\hat{\mu}N$ examples using the noises.*

*The model thus reaches zero training and manages to classify the large margin data at test. However, it fails to classify the small margin data because of the memorization step during training.*

**Why does GD+M generalize better than GD?** Since the large margin data are dominant, GD focus in priority on these examples to decrease its training loss. However, after fitting this data, it significantly lowers its gradient. The gradient is thus not large enough for learning the small margin data. Similarly, GD+M fits the large margin data and subsequently gets a small gradient. However,

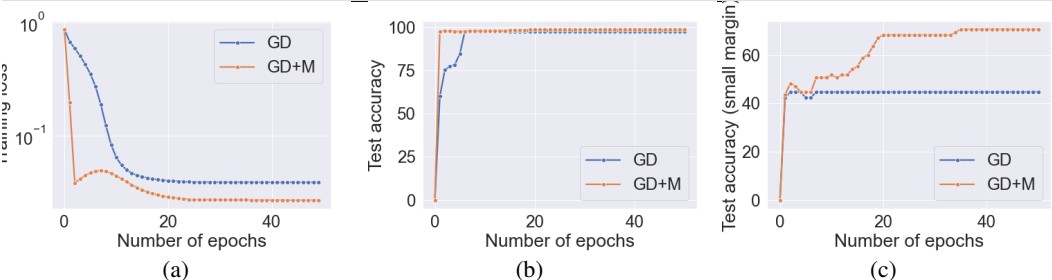

Figure 3: (a): Training loss (b) test accuracy on large margin data and (c) test accuracy on the small margin data in the synthetic setting. While GD and GD+M get zero training loss, GD+M generalizes better on small margin data than GD. Setting: 20000 training data, 2000 test data, d=30, number of neurons=5, number of patches=5.

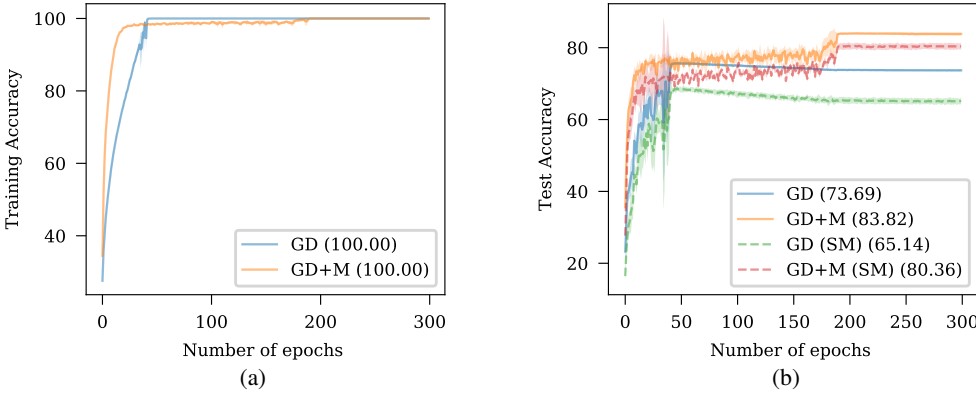

Figure 4: Training (a) and test (b) accuracy obtained with Resnet-18 on CIFAR-10 dataset with artificially generated small margin data. The architectures are trained using GD/GD+M for 300 epochs to ensure zero training error. Data augmentation, dropout and batch normalization are turned off. (SM) stands for the test accuracy obtained by the algorithm on the small margin data. Results are averaged over 5 runs with best scheduled learning rate and weight decay for each individual algorithm separately.

contrary to GD, GD+M has large historical gradients in his momentum gradient. These gradients helped to learn the feature in the large margin data. They also help to learn small margin data *since all the examples share the same feature*. GD+M therefore uses his momentum to learn the small margin data. We name this process *historical feature amplification* and believe that it is key to understand why momentum improves generalization.

**Empirical justification.** We also provide an empirical justification that such phenomenon does happen in a real-world setting as reported in Figure 4. In this experiment, we create small margin data in the CIFAR-10 dataset by respectively lowering the resolution of 10% of the training and test images, adding Gaussian noise of variance 0.005 and randomly shuffling the RGB channels. Figure 4 shows that even though both algortihms reach zero training error and 100% training accuracy, GD+M gets higher generalization than GD on this decimated dataset. Above all, at test, GD+M performs as well on small and large margin data while GD does relatively worse on small margin data.Indeed, the relative accuracy drop for GD+M is $80.36/83.32 = 0.97$ while for GD is equal to $65.14/73.69 = 0.88$.

Our paper is organized as follows. In Section 2, we formally define the data distribution equation (D), the model and algorithms we use to learn it. Lastly, Section 3 presents our main theorems and provide a proof sketch in Section 4 and Section 5. Additional experiments can be found in the Appendix.

MORE RELATED WORK

**Momentum in convex setting.** GD+M (a.k.a. heavy ball or Polyak momentum) consists in calculating the exponentially weighted average of the past gradients and using it to update the weights. For convex functions near a strict twice-differentiable minimum, GD+M is optimal regarding local convergence rate Polyak (1963; 1964); Nemirovskij & Yudin (1983); Nesterov (2003). However, it may fail to converge globally for general strongly convex twice-differentiable functions Lessard et al. (2015) and is no longer optimal for the class of smooth convex functions. In the stochastic setting, GD+M is more sensitive to noise in the gradients; that is, to preserve their improved convergence rates, significantly less noise is required d'Aspremont (2008); Schmidt et al. (2011); Devolder et al. (2014); Kidambi et al. (2018). Finally, other momentum methods are extensively used for convex functions such as Nesterov's accelerated gradient Nesterov (1983). Our paper focuses on the use of GD+M and contrary to the aforementioned papers, our setting is non-convex and we mainly focus on

the generalization of the model learned by GD and GD+M when both methods converge to global optimal. We underline that contrary to the non-convex world, generalization is typically disentangled with optimization for (strictly) convex functions.

**Non-convex optimization with momentum.** A long line of work consists in understanding the convergence speed of momentum methods when optimizing non-convex functions. Mai & Johansson (2020); Liu et al. (2020); Cutkosky & Mehta (2020); Defazio (2020) show that SGD+M reaches a stationary point as fast as SGD under diverse assumptions. Besides, Leclerc & Madry (2020) empirically shows that momentum accelerates neural network training for small learning rates and slows it down otherwise. Our paper differs from these works as we work in the batch setting and theoretically investigate the generalization benefits brought by momentum (and not the training ones).

**Generalization with momentum.** Momentum-based methods such as SGD+M, RMSProp (Tieleman & Hinton, 2012) and Adam (Kingma & Ba, 2014) are standard in deep learning training since the seminal work of Sutskever et al. (2013). Although its well accepted that Momentum improve generalization in deep learning, only a few works formally investigate the role of momentum in generalization. Leclerc & Madry (2020) *empirically* reports that momentum yields higher generalization when using a large learning rate. However, they assert that this benefit can be obtained by applying an even larger learning rate on vanilla SGD. We suspect that this observation is due to *batch normalization* (BN) which is known to dramatically bias the algorithm's generalization (Lyu & Li, 2019). In Appendix, we report that BN reduces the generalization gain of momentum comparing to without BN. To our knowledge, our work is first that *theoretically* investigate the generalization of momentum in deep learning.

## 2 SETTING AND ALGORITHMS

In this section, we first introduce a formal definition of the data distribution equation (D) and the neural network model we use to learn it. We finally present the GD and GD+M algorithms.

**General notations.** For a matrix $W \in \mathbb{R}^{m \times d}$, we denote by $w_r$ its $r$-th row. For a function $f : \mathbb{R}^{m \times d} \to \mathbb{R}$, we denote by $\nabla_{w_r} f(W)$ the gradient of $f$ with respect to $w_r$ and $\nabla f(W)$ the gradient with respect to $W$. For an optimization algorithm updating a vector $w$, $w^{(t)}$ represents its iterate at time $t$. We use $\mathbf{I}_d$ for the $d \times d$ identity matrix and $\mathbf{1}_m$ the all-ones vector of dimension $m$. Finally, we use the asymptotic complexity notations when defining the different constants in the paper. We use $\tilde{O}, \tilde{\Theta}, \tilde{\Omega}$ to hide logarithmic dependency on $d$.

**Data distribution.** We define our data distribution $\mathcal{D}$ as follows.

---

Each sample from $\mathcal{D}$ consists in an input data $X$ and a label $y$ that are generated as:

1. The label $y$ is uniformly sampled from $\{-1, 1\}$.

2. Each data-point $X = (X[1], \dots, X[P])$ consists in $P$ patches where each $X[j] \in \mathbb{R}^d$.

3. Signal patch: for one patch $P(X) \in [P]$, we have $X[P(X)] = cw^*$, where $c \in \mathbb{R}$, $w^* \in \mathbb{R}^d$ and $\|w^*\|_2 = 1$.            (D)

4. The distribution of $c$ satisfies that

$$c = \begin{cases} \alpha y & \text{with probability } 1 - \mu \\ \beta y & \text{with probability } \mu \end{cases}.$$

5. Noisy patches: for all the other patches $j \in [P] \backslash \{P(X)\}$, $X[j] \sim \mathcal{N}(0, (I - w^* w^{*\top})\sigma^2 \mathbf{I}_d)$.

---

We precise that we sample the noisy patches in the orthogonal complement of $w^*$ to have a simpler analysis. To present the simplest result, we assume that the values in equation (D) satisfy $\alpha = d^{0.49}$, $\beta = \frac{1}{\text{polylog(d)}\sqrt{d}}\alpha$, $\sigma = \frac{1}{\sqrt{d}}$ and $P \in [2, \text{polylog(d)}]$.

Using this model, we generate a training dataset $\mathcal{Z} = \{(X_i, y_i)\}_{i \in [N]}$ where $X_i = (X_i[j])_{j \in [P]}$. We focus on the case where $\mu = 1/\text{poly}(d)$ and $N = \Theta\left(\frac{1}{\mu}\right)$. We let $\mathcal{Z}$ to be partitioned in two sets $\mathcal{Z}_1$ and $\mathcal{Z}_2$ such that $\mathcal{Z}_1$ gathers the large margin data while $\mathcal{Z}_2$ the small margin ones. Lastly, we define $\hat{\mu} = \frac{|\mathcal{Z}_2|}{N}$ the fraction of small margin data.

**Learner model.** We use a two-layer convolutional neural network with cubic activation to learn the training dataset $\mathcal{Z}$. This model is the simplest non-linear network since a quadratic activation

would only output positive labels and mismatch our labeling function. The first layer weights are $W \in \mathbb{R}^{m \times d}$ and the second layer is fixed to $\mathbf{1}_m$. Given a input data $X$, the output of the model is

$$f_W(X) = \sum_{r=1}^{m} \sum_{j=1}^{P} \langle w_r, X[j] \rangle^3. \tag{CNN}$$

The number of neurons is set as $m = \mathrm{polylog}(d)$ to ensure that (CNN) is mildly over-parametrized.
**Training objective.** We fit the training dataset $\mathcal{Z}$ using (CNN) and solve the logistic regression problem

$$\min_{W \in \mathbb{R}^{m \times d}} \frac{1}{N} \sum_{i=1}^{N} \log\left(1 + \exp\left(-y_i f_W(X_i)\right)\right) + \frac{\lambda}{2} \|W\|_2^2 := \widehat{L}(W). \tag{P}$$

(P) sheds light on our choice of cubic activation in (CNN). Indeed, it is the smallest polynomial degree that makes the training objective (P) non-convex and compatible with our dataset. Linear or quadratic activations would respectively make the problem convex or all the labels positive. Here, we pick $\lambda \in \left[0, \frac{1}{\mathrm{poly}(d)N}\right]$.

**Importance of non-convexity.** When $\lambda > 0$, if the loss $\frac{1}{N} \sum_{i=1}^{N} \log\left(1 + \exp\left(-y_i f_W(X_i)\right)\right)$ is convex, then there is a unique global optimal solution, so the choice of optimization algorithm **does not matter**. In our case, due to the non-convexity of the training objective, GD + M converges to a different (approximate) global optimal comparing to GD, with better generalization properties.

**Test error.** We assess the quality of a predictor $\widehat{W}$ using the classical 0-1 loss used in binary classification. Given a sample $(X, y)$, the *individual test (classification) error* is defined as $\mathscr{L}(X, y) = \mathbf{1}\{f_{\widehat{W}}(X)y < 0\}$. While $\mathscr{L}$ measures the error of $f_W$ on an individual data-point, we are interested in the *test error* that measures the average loss over data points generated from ($\mathcal{D}$) and defined as

$$\mathscr{L}(f_{\widehat{W}}) := \mathbb{E}_{(X,y) \sim \mathcal{D}}[\mathscr{L}(f_{\widehat{W}}(X), y)]. \tag{TE}$$

**Algorithms.** We solve the training problem equation (P) using GD and GD+M. GD is defined by

$$W^{(t+1)} = W^{(t)} - \eta \nabla \widehat{L}(W^{(t)}), \ \ for \ t \geq 0, \tag{GD}$$

where $\eta > 0$ is the learning rate. On the other hand, GD+M is defined by the update rule

$$\begin{cases} g^{(t+1)} & = \gamma g^{(t)} + (1-\gamma)\nabla\widehat{L}(W^{(t)}) \\ W^{(t+1)} & = W^{(t)} - \eta g^{(t+1)} \end{cases}, \ \ for \ t \geq 0. \tag{GD+M}$$

where $\gamma \in (0, 1)$ is momentum factor. We now detail how to set parameters in (GD) and (GD+M).

**Parametrization 2.1.** *When running GD and GD+M on equation (P), the number of iterations is $T \in \left[\mathrm{poly(d)}N/(\eta), d^{O(\log d)}/(\eta)\right]$. For both algorithms, the weights $w_1^{(0)}, \ldots, w_m^{(0)}$ are initialized using independent samples from a normal distribution $\mathcal{N}(0, \sigma_0^2 \mathbf{I}_d)$ where $\sigma_0^2 = \frac{\mathrm{polylog}(d)}{d}$. The learning rate is set as:*

1. *GD: the learning rate may take any reasonable value $\eta \in (0, \tilde{O}(1)]$.*

2. *GD+M: the learning rate is a large learning rate: $\eta = \tilde{\Theta}(1)$.[1]*

*Lastly, the momentum factor in GD+M is set to be $\gamma = 1 - \frac{\mathrm{polylog}(d)}{d}$.*

Our Parametrization 2.1 matches with the parameters used in practice as the weights are generally initialized from Gaussian with small variance and momentum is set close to 1 (Sutskever et al., 2013).

## 3 MAIN RESULTS

We now formally state our main theorems regarding the generalization of models trained using equation (GD) and equation (GD+M) on the training set $\mathcal{Z}$ generated by equation ($\mathcal{D}$). As announced in the introduction, we show that the GD+M model incurs a generalization error that is dramatically smaller than the GD model. Before introducing the main result, we define some notations:

---

[1] This is consistent with the empirical observation that only momentum with large learning rate improves generalization (Sutskever et al., 2013)

**Main objects.** Let $r \in [m]$, $i \in [N]$, $j \in P \backslash \{P(X_i)\}$, $\gamma \in (0, 1)$ and $t \geq 0$. We are mainly interested in $w_r^{(t)}$, the $r$-th weight of the network, $\nabla_{w_r} \widehat{L}(W^{(t)})$ the gradient of the training loss w.r.t. $w_r$, $g_r^{(t)}$ the momentum gradient defined by $g_r^{(t+1)} = \gamma g_r^{(t)} + (1 - \gamma) \nabla_{w_r} \widehat{L}(W^{(t)})$. The analysis lies on the projection of these objects on the feature $w^*$ and on noisy patches $X_i[j]$. We introduce the following notations for the component of the learned weights along feature and noise directions:

– Projection on $w^*$: $c_r^{(t)} = \langle w_r^{(t)}, w^* \rangle$.

– Projection on $X_i[j]$ : $\Xi_{i,j,r}^{(t)} = \langle w_r^{(t)}, X_i[j] \rangle$.

– Total noise: $\Xi_i^{(t)} = \sum_{r=1}^m \sum_{j \in [P] \backslash \{P(X_i)\}} \langle w_r^{(t)}, X_i[j] \rangle^3$.

– Maximum signal: let $r_{\max} = \mathrm{argmax}_{r \in [m]} c_r^{(t)}$, $c^{(t)} = c_{r_{\max}}^{(t)}$

**Theorem 3.1.** *Assume that we run GD on (P) for $T$ iterations with parameters set as in Parametrization 2.1. With probability at least $1 - o(1)$, the weights learned by GD*

1. *partially learn the feature: for all $r \in [m]$, $|c_r^{(T)}| \leq \tilde{O}(1/\alpha)$.*

2. *memorize from small margin data: for all $i \in \mathcal{Z}_2$, $\Xi_i^{(t)} \geq \tilde{\Omega}(1)$.*

*Consequently, the training error is smaller than $O(\mu/\mathrm{poly}(d))$ and the test error is **at least** $\tilde{\Omega}(\mu)$.*

Intuitively, the training process of the GD model is described as follows. Since the large margin data are dominant in $\mathcal{Z}$, the gradient points mainly in the direction of the feature $w^*$. Therefore, GD eventually learns the feature in $\mathcal{Z}_1$ (Lemma 4.1) and the gradients from $\mathcal{Z}_1$ quickly become small. Afterwards, the gradient is dominated by the gradients from $\mathcal{Z}_2$ (Lemma 4.2). Because $\mathcal{Z}_2$ has small margin, the full gradient is now directed by the noisy patches. It implies that GD memorizes noise in $\mathcal{Z}_2$ (Lemma 4.4). Since these gradients also control the amount of remaining feature to be learned (Lemma 4.3), we conclude that the GD model partially learns the feature and introduces a huge noise component in the learned weights. We provide a proof sketch of Theorem 3.1 in Section 4.

**Theorem 3.2.** *Assume that we run GD+M on equation (P) for $T$ iterations with parameters set as in Parametrization 2.1. With probability at least $1 - o(1)$, the weights learned by GD+M*

1. *(at least for one of them) is highly correlated with the feature: $c^{(T)} > \tilde{\Omega}(1/\beta)$.*

2. *are barely correlated with noise: for all $r \in [m]$, for all $i \in [N]$ and $j \in [P]$. $|\Xi_{i,j,r}^{(T)}| \leq \tilde{O}(\sigma_0)$.*

*Consequently, the training loss and the test error are **at most** $O(\mu/\mathrm{poly}(d))$.*

Intuitively, the GD+M model follows this training process. Similarly to GD, it first fits the $\mathcal{Z}_1$ (Lemma 5.1). Contrary to GD, the momentum gradient is still highly correlated with $w^*$ after this step (Lemma 5.2). Indeed, the key difference is that momentum accumulates historical gradients. Since these gradients were accumulated when learning large margin data, the direction of momentum gradient is highly biased towards $w^*$. Therefore, the GD+M model *amplifies the feature* from these historical gradients to learn the feature in small margin data (Lemma 5.3). Subsequently, the gradient becomes small (Lemma 5.4) and the weights are no longer updated. Therefore, the GD+M model manages to ignore the noisy patches (Lemma 5.5) and learns the feature from both $\mathcal{Z}_1$ and $\mathcal{Z}_2$. We provide a proof sketch of Theorem 3.2 in Section 5.

To state the proof, we further decompose the gradients along signal and noise directions.

– Projection on $w^*$: $\mathscr{G}_r^{(t)} = \langle \nabla_{w_r} \widehat{L}(W_t), w^* \rangle$ and $\mathcal{G}_r^{(t)} = \langle g_r^{(t)}, w^* \rangle$.

– Projection on $X_i[j]$ : $\mathsf{G}_{i,j,r}^{(t)} = \langle \nabla_{w_r} \widehat{L}(W^{(t)}), X_i[j] \rangle$, $G_{i,j,r}^{(t)} = \langle g_r^{(t)}, X_i[j] \rangle$.

– Maximum signal: let $r_{\max} = \mathrm{argmax}_{r \in [m]} c_r^{(t)}$, $c^{(t)} = c_{r_{\max}}^{(t)}$ and $\mathcal{G}^{(t)} = \mathcal{G}_{r_{\max}}^{(t)}$.

**Signal and noise iterates.** Our analysis is build upon a decomposition of the updates equation (GD) and equation (GD+M) on $w^*$ and $X_i[j]$. These decompositions are respectively defined as follows:

$$c_r^{(t+1)} = c_r^{(t)} - \eta \mathscr{G}_r^{(t)} \quad \text{(GD-S)} \qquad \qquad \Xi_{i,j,r}^{(t+1)} = \Xi_{i,j,r}^{(t)} - \eta \mathsf{G}_{i,j,r}^{(t)} \quad \text{(GD-N)}$$

$$
\begin{cases}
\mathcal{G}_r^{(t+1)} = \gamma \mathcal{G}_r^{(t)} + (1-\gamma)\mathscr{G}_r^{(t)} \\
c_r^{(t+1)} = c_r^{(t)} - \eta \mathcal{G}_r^{(t+1)}
\end{cases} \text{(GDM-S)}
\qquad
\begin{cases}
G_{i,j,r}^{(t+1)} = \gamma G_{i,j,r}^{(t)} + (1-\gamma)\mathsf{G}_{i,j,r}^{(t)} \\
\Xi_{i,j,r}^{(t+1)} = \Xi_{i,j,r}^{(t)} - G_{i,j,r}^{(t+1)}
\end{cases}
$$
$$
\text{(GDM-N)}
$$

We detail how to use these dynamics to analyze GD+M and GD in Section 4 and Section 5. Our analysis heavily depends on the gradients of the training loss which involve $\text{sigmoid}(x) = (1 + e^{-x})^{-1}$. We define the derivative of a data-point $i$ as $\ell_i^{(t)} = \text{sigmoid}(-y_i f_{W^{(t)}}(X_i))$, the derivatives $\nu_k^{(t)} = \frac{1}{N}\sum_{i \in \mathcal{Z}_k} \ell_i^{(t)}$ for $k \in \{1, 2\}$ and the full derivative $\nu^{(t)} = \nu_1^{(t)} + \nu_2^{(t)}$.

## 4 ANALYSIS OF GD

In this section, we provide a proof sketch for Theorem 3.1 that reflects the behavior of GD with $\lambda = 0$. A more detailed proof (extending to $\lambda > 0$) can be found in the Appendix.

**Step 1: Learning $\mathcal{Z}_1$.** At the beginning of the learning process, the gradient is mostly dominated by the gradients coming from the $\mathcal{Z}_1$ samples. Since these data have large margin, the gradient is thus highly correlated with $w^*$ and $c_r^{(t)}$ increases as shown in the following Lemma.

**Lemma 4.1.** *For all $r \in [m]$ and $t \geq 0$, equation (GD-S) is simplified as:*

$$
c_r^{(t+1)} \geq c_r^{(t)} + \tilde{\Theta}(\eta)\alpha^3 (c_r^{(t)})^2 \cdot \text{sigmoid}(-\textstyle\sum_{s=1}^{t} \alpha^3 (c_s^{(t)})^3).
$$

*Consequently, after $T_0 = \tilde{\Theta}\left(\frac{1}{\eta \alpha^3 \sigma_0}\right)$ iterations, for all $t \in [T_0, T]$, we have $c^{(t)} \geq \tilde{\Omega}(1/\alpha)$.*

Intuitively, the increment in the update in Lemma 4.1 is non-zero when the sigmoid is not too small which is equivalent to $c^{(t)} \leq \tilde{O}(1/\alpha)$. Therefore, $c^{(t)}$ keeps increasing until reaching this threshold. After this step, the $\mathcal{Z}_1$ data have small gradient and therefore, GD has learned these data.

**Lemma 4.2.** *Let $T_0 = \tilde{\Theta}\left(\frac{1}{\eta \alpha^3 \sigma_0}\right)$. After $t \in [T_0, T]$ iterations, the $\mathcal{Z}_1$ derivative is bounded as* $\nu_1^{(t)} \leq \tilde{O}\left(\frac{1}{\eta(t-T_0+1)\alpha}\right) + \tilde{O}\left(\frac{\beta^3}{\alpha}\right)\nu_2^{(t)}$. *The full derivative is $\nu^{(t)} \leq \tilde{O}\left(\frac{1}{\eta(t-T_0+1)\alpha} + \left(1 + \frac{\beta^3}{\alpha}\right)\nu_2^{(t)}\right)$.*

By our choice of parameter, Lemma 4.2 indicates that the full gradient is dominated by the gradients from $\mathcal{Z}_2$ data after $T_0 = \tilde{\Omega}\left(\frac{1}{\tilde{\mu}\eta\alpha}\right)$. Consequently, $\nu_2^{(t)}$ also rules the amount of feature learnt by GD.

**Lemma 4.3.** *Let $T_0 = \tilde{\Theta}\left(\frac{1}{\eta \alpha^3 \sigma_0}\right)$. For $t \in [T_0, T]$, equation (GD-S) becomes $c^{(t+1)} \leq \tilde{O}(1/\alpha) + \tilde{O}(\eta\beta^3/\alpha)\sum_{\tau=T_0}^{t} \nu_2^{(\tau)}$.*

Lemma 4.3 implies that quantifying the decrease rate of $\nu_2^{(t)}$ provides an estimate on the quantity of feature learnt by the model. We remark that $\nu_2^{(t)} = \text{sigmoid}(\beta^3 \sum_{s=1}^{m}(c_s^{(t)})^3 + \Xi_i^{(t)})$ for some $i \in \mathcal{Z}_2$. We thus need to determine whether the feature or the noise terms dominates in the sigmoid.

**Step 2: Memorizing $\mathcal{Z}_2$.** We now show that the total correlation between the weights and the noise in $\mathcal{Z}_2$ data increases until being large.

**Lemma 4.4.** *Let $t \geq 0$ and $i \in \mathcal{Z}_2$. Assume that $\Xi_i^{(t)} \leq \tilde{O}(1)$. Then, equation (GD-N) can be simplified as:*

$$
y_i \Xi_{i,j,r}^{(t+1)} \geq y_i \Xi_{i,j,r}^{(0)} + \frac{\tilde{\Theta}(\eta\sigma^2 d)}{N}\sum_{\tau=0}^{t}(\Xi_{i,j,r}^{(\tau)})^2 - \tilde{O}\left(\frac{P\sigma^2\sqrt{d}}{\alpha}\right).
$$

*Let $T_1 = \tilde{O}\left(\frac{N}{\sigma_0 \sigma\sqrt{d}\sigma^2 d}\right)$. Therefore, $\Xi_i^{(t)} \geq \tilde{\Omega}(1)$, for $t \in [T_1, T]$ and thus GD memorizes.*

By Lemma 4.4, in the gradient of $\mathcal{Z}_2$ data, the noise term dominates the feature term (which scales as $\tilde{O}(\beta^3)$). Consequently, the algorithm memorizes the $\mathcal{Z}_2$ data which implies a fast decay of $\nu_2^{(t)}$.

**Lemma 4.5.** *Let $T_1 = \tilde{O}\left(\frac{N}{\sigma_0 \sigma \sqrt{d}\sigma^2 d}\right)$. For $t \in [T_1, T]$, we have $\sum_{\tau=0}^{t} \nu_2^{(\tau)} \leq \tilde{O}\left(\frac{1}{\eta \sigma_0}\right)$.*

Combining Lemma 4.5 and Lemma 4.3, we prove that GD partially learns the feature.

**Lemma 4.6.** *For $t \leq T$, the signal component satisfies $c^{(t)} \leq \tilde{O}(1/\alpha)$.*

Lemma 4.4 and Lemma 4.6 respectively yield the first two items in Theorem 3.1. Bounds on the training and test errors are respectively obtained by plugging these results in (P) and (TE).

## 5 ANALYSIS OF GD+M

In this section, we provide a proof sketch for Theorem 3.2 that reflects the behavior of GD+M with $\lambda = 0$. A more detailed proof (also extending to $\lambda > 0$) can be found in the Appendix.

**Step 1: Learning $\mathcal{Z}_1$.** Similarly to GD, by our initialization choice, the early gradients and so, momentum gradients are large. They are also spanned by the feature $w^*$ and therefore, the GD+M model also increases its correlation with $w^*$.

**Lemma 5.1.** *For all $r \in [m]$ and $t \geq 0$, as long as $c^{(t)} \leq \tilde{O}(1/\alpha)$, the momentum update equation (GDM-S) is simplified as:*
$$-\mathcal{G}_r^{(t+1)} = -\gamma \mathcal{G}_r^{(t)} + (1-\gamma)\Theta(\alpha^3)(c_r^{(t)})^2$$
*Consequently, after $\mathcal{T}_0 = \tilde{\Theta}\left(\frac{1}{\sigma_0 \alpha^2} + \frac{1}{1-\gamma}\right)$ iterations, for all $t \in [\mathcal{T}_0, T]$, we have $c^{(t)} \geq \tilde{\Omega}(1/\alpha)$.*

**Step 2: Learning $\mathcal{Z}_2$.** Contrary to GD, GD+M has a large momentum that contains $w^*$ after Step 1.

**Lemma 5.2.** *Let $\mathcal{T}_0 = \tilde{\Theta}\left(\frac{1}{\sigma_0 \alpha^3} + \frac{1}{1-\gamma}\right)$. For $t \in [\mathcal{T}_0, T]$, we have $\mathcal{G}^{(t)} \geq \tilde{\Omega}(\sqrt{1-\gamma}/\alpha)$.*

Lemma 5.2 hints an important distinction between GD and GD+M: while the current gradient along $w^*$ is small at time $\mathcal{T}_0$, the momentum gradient stores historical gradients that are spanned by $w^*$. It *amplifies* the feature present in previous gradients to learn the feature from small margin data.

**Lemma 5.3.** *Let $\mathcal{T}_0 = \tilde{\Theta}\left(\frac{1}{\sigma_0 \alpha^3} + \frac{1}{1-\gamma}\right)$. After $\mathcal{T}_1 = \mathcal{T}_0 + \tilde{\Theta}\left(\frac{1}{1-\gamma}\right)$ iterations, for $t \in [\mathcal{T}_1, T]$, we have $c^{(t)} \geq \tilde{\Omega}\left(\frac{1}{\sqrt{1-\gamma}\alpha}\right)$. Our choice of parameter in Section 2, this implies $c^{(t)} \geq \tilde{\Omega}(1/\beta)$.*

Lemma 5.3 states that at least one of the weights that is highly correlated with the feature compared to GD where $c^{(t)} = \tilde{O}(1)$. This result implies that $\nu^{(t)}$ converges fast.

**Lemma 5.4.** *Let $\mathcal{T}_0 = \tilde{\Theta}\left(\frac{1}{\eta \sigma_0 \alpha^3} + \frac{1}{1-\gamma}\right)$. After $\mathcal{T}_1 = \mathcal{T}_0 + \tilde{\Theta}\left(\frac{1}{1-\gamma}\right)$ iterations, for $t \in [\mathcal{T}_1, T]$, $\nu^{(t)} \leq \tilde{O}\left(\frac{1}{\eta(t-\mathcal{T}_1+1)\beta}\right)$.*

With this fast convergence, Lemma 5.4 implies that the correlation of the weights with the noisy patches does not have enough time to increase and thus, remains small.

**Lemma 5.5.** *Let $i \in [N]$, $j \in [P] \backslash \{P(X_i)\}$ and $r \in [m]$. For $t \geq 0$, equation (GDM-N) can be rewritten as $|G_{i,j,r}^{(t+1)}| \leq \gamma |G_{i,j,r}^{(t)}| + (1-\gamma)\tilde{O}(\sigma_0^2 \sigma^4 d^2)\nu^{(t)}$. As a consequence, after $t \in [\mathcal{T}_1, T]$ iterations, we thus have $|\Xi_{i,j,r}^{(t)}| \leq \tilde{O}(\sigma_0 \sigma \sqrt{d})$.*

Lemma 5.3 and Lemma 5.5 respectively yield the two first items in Theorem 3.2.

## 6 DISCUSSION

Our work is a first step towards understanding the algorithmic regularization of momentum and leaves room for improvements. We constructed a data distribution where historical feature amplification may explain the generalization improvement of momentum. However, it would be interesting to understand whether this phenomenon is the only reason or whether there are other mechanisms explaining momentum's benefits. An interesting setting for this question is NLP where momentum is used to train large models as BERT (Devlin et al., 2018). Lastly, our analysis is in the batch setting to isolate the generalization induced by momentum. It would be interesting to understand how the stochastic noise and the momentum together contribute to the generalization of a neural network.

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
