# OpenReview forum: "Towards understanding how momentum improves generalization in deep learning"
_ICLR.cc/2022/Conference — ICLR 2022 Submitted_

### Official Review · Reviewer_6raf · 2021-10-19

**Correctness:** 2
**Technical Novelty And Significance:** 3
**Empirical Novelty And Significance:** 3
**Recommendation:** 5
**Confidence:** 4

**Main Review:**

This paper shows that gradient descent + momentum (a.k.a. Heavy Ball) provably generalizes better than standard gradient descent for training a two-layer convolution network when the data distribution has a particular structure. Specifically, it considers a binary classification dataset of size N of which $(1-\hat{u}) N$ are with a large margin and $\hat{\mu} N$ are with a small margin. Most of the data points are with a large margin, i.e. $\hat{u} << 1$

The paper starts by giving some empirical evidence --- GD+M generalizes better for training Resnet-18 and WideResnet on CIFAR-10 and CIFAR 100 than GD. The paper also reports an experiment on a synthetic dataset in which features are from a Guassian distribution and shows that momentum does not improve over GD, which suggests that whether momentum has an advantage over GD depends on the underlying data distribution.

As the authors argue, the key to showing that momentum generalizes better is by showing that after a certain number of iterations, even though the current gradient could be small for those data points with a small margin, the momentum (which is an weighted average of previous gradients) has a sufficient size of projection onto the signal direction $w_*$. Hence, GD+M can still learn $w_*$ from the small margin data.

However, the proof in the appendix in my opinion is not written well. There are a few typos and a lot of approximations, e.g. a lot of $O(\cdot)$ and $\Theta(\cdot)$. For some of the approximations, it is hard to tell if the approximations make sense.  Here are some questions regarding the analysis.


1. (Lemma H.5): Can the authors provide a detailed derivation of the inequality (174) from (173)? The authors claim that by using Lemma H.6 one can get (174) but it is not clear at all from my perspective. Specifically, Lemma H.6 writes

$$\hat{L}(W^{(t)}) = \Theta(1) \left( \hat{L}^{(t)}(\alpha) +  \hat{L}^{(t)}(\beta)    \right)$$

$$\hat{L}(W^{(t)} - \eta \gamma g^{(t)} ) \leq \Theta(1) \left( \hat{L}^{(t)}(\alpha) +  \hat{L}^{(t)}(\beta)    \right)$$

Did the authors try to show $\hat{L}(W^{(t)} - \eta \gamma g^{(t)} ) \leq \hat{L}(W^{(t)})$? If so, why don't just prove $\hat{L}(W^{(t)} - \eta \gamma g^{(t)} ) \leq \hat{L}(W^{(t)})$ directly? Is this used anywhere in the proof of Lemma H.5?
Can the authors write down precisely the constant that $\Theta(1)$ in (180) and (181) represents?
Furthermore, can the authors comment on why the approximation (188) makes sense?
Also, can the authors expand on the last sentence of the proof? How to apply Lemma I.24 to get (181)?

2. (Lemma H.5): The first line on Page 48 says "by Induction hypothesis C.2, we know that $ c_r^{(t)} \geq 0$ for all t".
But from Induction hypothesis C.2 (Page 16), it is only shown that $c_r^{(t)}$ is larger than a negative number. The validity of Lemma H.5 is concerning given this and the above concern.


3. (Lemma I.15): The authors show that $z_t \geq v$ for all $t \geq t_0$. Can the authors explain where exactly in the proof did they show the "for all $t \geq t_0$" part?

A related question about Lemma I.16-I.17: Should one also show $z_t \geq v$ for all $t \geq t_0$ like Lemma I.15?

4. (Lemma 4.4): Can the authors elaborate on "Therefore, the sequence $y_i \Xi_{i,j,r}^{(t)}$ is non-decreasing" based on the inequality in the lemma?

5. (Lemma 5.1~5.4) We see that the iterate complexities in these lemmas all have a $\frac{1}{1-\gamma}$ factor. Does it suggest that the progress of GD+momentum is slower when a larger value of the momentum parameter $\gamma$ is used?

6. (Lemma 5.2) The lemma shows the size of momentum is lower bounded as $g^{(t)} \geq O( \sqrt{1-\gamma} )$. The lower bound becomes smaller and smaller as the momentum parameter $\gamma$ increases. I found this counter-intuitive. The authors might want to explain this.

7. (The sentence right below Lemma 5.4) QUOTE With this fast convergence, Lemma 5.4 implies that the weight of... are barely update. UNQUOTE
The authors might want to expand on how Lemma 5.4 implies this.

8. (Typo on B.2) Check the definition of $r_{\max}$

9. (Typo on (4)) $y_i$ is dropped.

10. Sentence right after (8) is not complete.

11. ((65) on Page 9) Can the authors provide a detailed derivation of the inequalities on (65)?

12.  Regarding the intuition of why GD+M generalizes better than GD --- the authors claim that it is because momentum stores historical gradients that are spanned by w_* and hence amplifies the features present in the previous gradients and helps learn the feature from the small margin data. Can the authors quantize how large is the signal in the momentum term compared to the signal in the gradient term? Is it $\frac{1}{1-\gamma}$ larger? This is not transparent from the proof.


**Summary Of The Paper:**

This paper shows that gradient descent + momentum (a.k.a. Heavy Ball) provably generalizes better than standard gradient descent for training a two-layer convolution network when the data distribution has a particular structure.

**Summary Of The Review:**

The analysis is not very clear, a lot of approximations are made throughout the proof, and some places in the proof are concerning. I don't recommend an acceptance at the current point.

---

> ### Author Response · Authors · 2021-11-23
> **Response to reviewer 6raf**
>
> We would like thank the reviewer 6raf for  all the comments and pointers to the typos in the proof. We acknowledge that during the submission there were some not-so-well-stated statements in the technical lemmas. During the revision period, we revised parts of the proof and believe that the proof is now in good shape. We point out that the corrections do not change the main lemmas/theorems of the paper. Lastly, we would like to underline that the review is  of **excellent quality** and would like to highlight the amazing work of the reviewer. Their comments helped us to significantly improve the paper.
>
> **-- Previous Lemma H.5:  $\widehat{L}(W^{(t)})=\Theta(1)((1-\mu)\hat{\mathcal{L}}^{(t)}(\alpha)+\mu\hat{\mathcal{L}}^{(t)}(\beta))$ and $\hat{L}(W^{(t)}-\eta\gamma g^{(t)})\leq \Theta(1)((1-\mu)\hat{\mathcal{L}}^{(t)}(\alpha)+\mu\hat{\mathcal{L}}^{(t)}(\beta))$.**
>
>  The previous Lemma H.5. had some misleading part. Therefore, we update a new  proof the convergence rate of the loss function when using GD+M (see Lemma I.5.). The proof is now by contradiction and does not require any descent lemma.
>
> **-- Previous Lemma I.15: where exactly in the proof did they show the "for all $t\geq t_0$ part?**
>
> This Lemma is now Lemma J.15. The sequence is non-decreasing because $z^{(t+1)}-z^{(t)}\geq +m(z^{(t)})^2\geq 0.$ Therefore, there is a time where $z^{(t)}\geq\upsilon.$
>
> **-- Previous Lemma I.16 (Now J.16):  show for all $z_t\geq \upsilon$ for all $t\geq t_0$ like Lemma I.15?**
>
>  Indeed, in this case it is less obvious that there exists such a time because the sequence $z_t$ is not strictly non-decreasing this time. What we have is $C \leq o(z_0)$  so $z_t \geq z_s(1 - o(1))$ for every $s \leq t$.  The proof actually proves that such time where  $z_t\geq \upsilon$ exists. The intuition is that when $t$ is very large, the term $A\sum_{\tau=0}^t z_{\tau}^2$ will dominate and therefore help to increase the sequence. But this is only after a very large time.
>
> **-- Lemma 4.4: "Therefore, the sequence is non-decreasing".**
>
>  Indeed, this is incorrect. The sequence is ``approximately" non-decreasing as explaining in the previous points regarding Lemma I.16. We accidentally missed the word approximately. We corrected this in the revised version of the paper.
>
>
>  **--  The lower bound becomes smaller and smaller as the momentum parameter $\gamma$ increases.**
>
>  Indeed, the momentum is smaller as $\gamma$ increases, however, due to the update of momentum, it also takes $\frac{1}{1 - \gamma} $ iterations for some component to be removed from the momentum. Overall, the total contribution (accross all iterations) is $\sqrt{1 - \gamma} \times \frac{1}{1 - \gamma}$ which is an increasing function with $\gamma$.
>
> **-- Expand on how Lemma 5.4 implies weight of... are barely update.**
>
> We agree that this sentence may be ambiguous. We corrected it in the revised paper. What we meant is that with this fast convergence of the gradients $\nu^{(t)}$,  the correlation of GD+M's weights with the noisy patches does not have enough time to increase and thus, remains small.
>
> **-- Detailed explanation on equation 65 (in old paper).**
>
> This is just the momentum update $\mathcal{G}^{(t)}=\gamma\mathcal{G}^{(t-1)}-\eta\mathscr{G}^{(t)}$. Since we are at early iterations and have small initialization, the sigmoid terms are still non-zero i.e. $\Theta(1)\leq \hat{\ell}^{(t)}(\alpha),\hat{\ell}^{(t)}(\beta)\leq\Theta(1)$. Therefore, one has $\eta\mathscr{G}^{(t)}\leq \Theta(1) [(1-\mu)\alpha^3+\mu\beta^3]c_t^2 .$
>
>
> **-- how large is the signal in the momentum term compared to the signal in the gradient term?**
>
> In Induction hypothesis C.2, we proved that the signal in GD $\leq \tilde{O}(1/\alpha)$. On the other hand, we proved in Lemma 5.3 the signal for GD+M is $\geq \tilde{\Omega}(1/\sqrt{1-\gamma}\alpha).$ Therefore, as the reviewer said, the gap is at least $\tilde{\Omega}(1/\sqrt{1-\gamma})$.

---

### Official Review · Reviewer_PBhC · 2021-10-31

**Correctness:** 4
**Technical Novelty And Significance:** 4
**Empirical Novelty And Significance:** 4
**Recommendation:** 6
**Confidence:** 2

**Main Review:**

I am not an expert in this area and am not very familiar with the results presented by this paper.  But after I search and read related papers, I think this paper is very novel.

1. Does momentum unconditionally improve generalization in deep learning is quite important in ML and AI. This paper answers this problem negatively by a construction method.

2. This paper presents an insight that momentum helps to learn small margin data since all the examples share the same feature. I think this idea is wonderful and novel.

3. This paper provides plenty of detailed theoretical results. Many technical lemmas are developed.

4. The authors present the corresponding numerical experiments to demonstrate the theoretical findings.

**Summary Of The Paper:**

This paper shows that the momentum conditionally improves the generalization by reconstructing a data case.

**Summary Of The Review:**

I do not think this paper is below the accept bar due to its interesting topic, novel analysisis, interesting result, complicated techniques, and good presentation.

---

### Official Review · Reviewer_Jrkp · 2021-11-01

**Correctness:** 2
**Technical Novelty And Significance:** 3
**Empirical Novelty And Significance:** 2
**Recommendation:** 5
**Confidence:** 4

**Main Review:**

This paper considers a very important optimization problem in real application especially deep learning. The paper is well-organized and very easy to understand. The authors provided a very solid theoretical results under a very special case and some assumptions. However, I still have some questions to be addressed.

1. Due to the theoretical results of GD and GD+M are built on a very special case, there is a big gap between the results in this paper and deep learning. It seems that the authors over-claimed the contribution of this paper.

2. The authors claimed that they only focused on GD+M since the empirical results of GD+M can generalize better than GD (e.g., Figure 1). However, this is not realistic since the authors did not conduct GD+M and GD but large mini-batching SGD+M and SGD. In fact, the results only show that large mini-batching SGD+M can generalize better than large mini-batching SGD. Therefore, it is reasonable to focus on the contribution of momentum of the large mini-batching stochastic gradient (rather than "true gradient") on generalization.

3. It is strongly recommended and needed to provide experiments to verify the theoretical result. At least, the authors should provide numerical results on a simulation data according to the data distribution $\mathcal D$ (page 5) and the learning problem (page 6). The numerical results will make the contribution of this paper complete and solid.

**Summary Of The Paper:**

Through some experiments, this paper claims that momentum does not always lead to a higher generalization in deep learning and such benefit seems to heavily depend on both the structure of the data and the learning problem. Then, the authors considered a certain data structure (large and small margin data) and learning problem (binary classification problem with 2-layer CNN model and logistic loss). Under this special case, the authors shown that both GD and GDM reach zero training error and perfectly classify large margin data, but GD fails to classify the small margin data while GDM can still perfect classify small margin data due to the historical gradients caused by the momentum.

**Summary Of The Review:**

1. Solid theoretical results under a special case

2. Big gap in the paper

3. The reason of considering GD+M

4. Lack of experiment (for verifying the theoretical results)

---

> ### Author Response · Authors · 2021-11-23
> **Response to reviewer Jrkp**
>
> We thank the reviewer Jrkp for their review. We would like to clarify some points in the paper to ensure that the purpose of the paper is well-understood.
>
> **-- "big gap between the results in this paper and deep learning."**
>
> As the title of the paper indicates, our work is a first step towards understanding how momentum improves generalization in deep learning. We are not aware of any work that understands this phenomenon in deep learning. We would like to emphasize that the theoretical setting we consider can be seen a mathematical modelling of what we have in image classification:
>
> * **dataset**: as we explained in the paper, in a image classification dataset such as CIFAR-10, we have patches that are informative (like the patch which contains the nose of a dog) and others that are not (like the patches containing the background). Besides, in some images in CIFAR-10, it is easy to recognize the category of these images thanks to these informative patches and for others, it is hard because the informative patch is blurry. Our dataset is a mathematical translation of this observation. Easy-to-recognize images are large margin data and hard-to-recognize images are small margin data.
>
> * **model**: neural networks with polynomial activation is well-motivated in theory community.   A  lot  of  prior  theoretical  works  focus  on  neural  networks  with  quadratic/cubic activations ([1,2,3]  among others), and they are known to perform comparably with neural networks using ReLU activations [3].  Moreover, for a cubic activation function,a(<w,x >)3=< a1/3w,x >3.  Therefore, training the second layer is not needed (can be absorbed in the hidden layer) so we simply set them to be one.
>
> * **batch-size**: reasonable to focus on the contribution of momentum of the large mini-batching stochastic gradient (rather than "true gradient") on generalization.
>
> **-- Experiments with batch size=1024.**
>
> As stated in the paper, when the batch size is 1024, it has been observed that the stochastic gradient concentrates around the true gradient [4]. Therefore, the experiment in Figure 1 is meant to simulate GD vs SGD. Nevertheless, we have updated Figure 1 in the submission using  **full batch** GD . What we observed is that GD+M gets 84.68% (against 85.20% when the batch size was 1024) and GD gets 75.83% (against 76.27%).  The test accuracies are approximately similar and therefore, the conclusions are the same as the ones in the paper.
>
> Moreover, in the appendix (both in the original paper and in the revised version), we show that increasing the batch-size leads to a BIGGER generalization gap between SGD+M and SGD, indicating that momentum is more useful in the larger batch-size regime.
>
> **-- "experiments to verify the theoretical result".**
>
> We thank the reviewer for this suggestion and fully agree with them. We have run additional experiments  (see Figure 3). We see that the GD+M model is able to better-classify the small margin data compared to GD, while on large margin data they behave identically.
>
>
>
> [1] Li, Y., et al. Algorithmic regularization in over-parameterized matrix sensing and neural networks with quadratic activations. 2018.
>
> [2] Woodworth, B., et al. Kernel and rich regimes in overparametrized models. 2020.
>
> [3] Allen-Zhu, Z., et al. Backward feature correction: How deep learning performs deep learning. 2020.
>
> [4] Cohen, Jeremy M., et al. Gradient descent on neural networks typically occurs at the edge of stability. 2021.

---

### Official Review · Reviewer_osph · 2021-11-02

**Correctness:** 3
**Technical Novelty And Significance:** 2
**Empirical Novelty And Significance:** 2
**Recommendation:** 3
**Confidence:** 4

**Main Review:**

The main strength of this paper is a rigorous mathematical analysis that exhaustively justifies the generalization gains from adding momentum to the gradient descent algorithm. The analysis is solid and sound (although I did not manage to check all the details carefully) and could be interesting from a mathematical perspective.

However, these results reflect only a specific case of problem statement, which is the major weakness of this submission. Namely, the authors consider a binary classification problem with a two-layer convolutional neural network with cubic activation and a fixed second layer. The data is generated according to a specific scheme where each data point has a single patch containing useful information (the signal patch), and all the others are Gaussian random noise with a small variance; the data is split into large-margin and small-margin parts according to the intensity of the signal patch. The authors claim that real-world datasets possess similar properties, however, they do not provide sufficient grounding for that argument. Figure 3 exhibits that momentum indeed gives more gains if real data are artificially augmented with small margin data (which accords with the provided theory), but it does not prove that real data initially *had* such structure.
In other words, even if the derived theory is correct in the particular case, it still does not suffice to explain the benefits of momentum in deep learning unless the authors prove that the considered case is relevant for practical deep learning. That becomes especially acute in light of the provided counterexample with teacher-student learning on Gaussian data. It is unclear why the authors consider their setup more relevant to deep learning than the latter.

Other concerns:
1. "Indeed, a batch size of 1024 is known to be large enough in CIFAR training to consider the stochastic gradient relatively close to the full gradient (Cohen et al., 2021)". I could not find any reference in Cohen et al. (2021) about this. In fact, Cohen et al. (2021) train their models on subsets of original datasets to perform actual full-batch gradient descent. I am not convinced that taking a batch size of 1024 in SGD reduces it to GD. Moreover, such large-batch settings are well studied in prior work (see, e.g., https://arxiv.org/abs/2006.15081) and are known as "curvature dominated
regime", in which SGD with momentum typically outperforms vanilla SGD. I suggest the authors reconduct their experiments in the proper full-batch GD setting.
2. "In this paper, we ... empirically show that gradient descent with momentum (GD+M) significantly improves generalization comparing to gradient descent (GD) in many deep learning tasks". In the main body of the paper, I could only find experiments with ResNet and Wide-ResNet on CIFAR datasets, which does not seem like "many deep learning tasks". I suggest either providing more experiments or softening the wording.
3. The authors did not provide details on their experimental setup.

Minor concerns:
* The results are stated for the momentum with updates of type $g^{(t+1)} = \gamma g^{(t)} + (1 - \gamma) \nabla \hat{L}(W^{(t)})$ while the conventional momentum update takes the form of $g^{(t+1)} = \gamma g^{(t)} + \nabla \hat{L}(W^{(t)})$. How do the results carry over to this case?
* Table 3(b) in supplementary demonstrates that for large batch sizes GD+M performs worse than conventional GD. Is that a typo or an outlier?
* "An interesting setting for this question is NLP where momentum is used to train large models as BERT (Devlin et al., 2018)". As far as I know, large NLP models are usually trained with Adam or more sophisticated optimizers rather than SGD+M.
* "We suspect that this observation is due to batch normalization (BN) which is known to dramatically bias the algorithm’s generalization (Lyu & Li,
2019)". To my knowledge, Lyu & Li (2019) did not write about BN, please, correct the citation.

---

**UPD**
After reading the authors' response, I am inclined to keep my score unchanged.

**Summary Of The Paper:**


This paper investigates the generalization benefits of using momentum when training neural networks with gradient descent. In contrast with existing literature, which studies momentum mostly empirically and in the stochastic setting, the authors develop a theoretical explanation for why generalization improves when optimization is performed using full-batch gradient descent with momentum (GD+M) than without it (GD). Their analysis focuses on a specific structure of the data and the learning problem, and the authors argue that similar assumptions apply to the real-world datasets used in practice. The authors prove two main results: one about GD learning large-margin data and overfitting to small-margin data and the other about GD+M successfully learning all the data thanks to historical gradients. Interestingly, the authors provide a counterexample where momentum does not aid generalization (or even worsen it), which offers good food for thought.

**Summary Of The Review:**

This paper provides a solid theoretical analysis, which, however, applies only to a specific problem statement and thus cannot be considered a satisfactory argument for why momentum is beneficial in deep learning (unless the authors provide more evidence that their setting is relevant to real-world datasets). The empirical justification does not correctly reflect the considered full-batch gradient descent training. I believe the paper could benefit from another round of revision and is not ready for publication yet.

---

> ### Author Response · Authors · 2021-11-23
> **Response to Reviewer osph**
>
> We thank the reviewer osph for their review.
>
> -- specific case of problem statement, which is the major weakness of this submission.
>
> We would like to insist that momentum does not unconditionally improve generalization. Table 1 shows that when the data is Gaussian, the models trained with GD and GD+M have approximately the same test accuracy. Therefore, we needed to find a dataset where momentum improves generalization. Our explanation is that "if a data set consists of small-large margin data", then "the GD+M model learns the small margin data better compare to the GD one", we designed the most simple dataset that matches this hypothesis. Regarding the learning problem, we focus on the 2-layer CNN which is, **to the best of our knowledge, all the prior works (cited in our paper, and ([1,2,3] among others)) on theory aspects of algorithmic regularization considers 2-layer neural networks**. Please note that polynomial activations are popular in the deep learning theory community (see [1,2,3] among others). Thus, a simplified setting is necessary to characterize the mechanism behind why momentum may improve generalization and for what type of datasets.
>
> Finally, we would like to highlight that similarly to us, essentially all implicit bias results assume a certain structure of the dataset/architecture  and also only works under two-layer neural networks. Without these structural assumptions and simplified neural networks, it is impossible to prove such a result at the current stage of theory development.
>
> **We politely ask the reviewer to provide one example of a paper in algorithmic regularization literature (where one PROVES algorithm learns solutions that generalize better than the other), so the reviewer considers it as not "too specific"**
>
> We insist that the point of the paper is not to make a general statement on the generalization improvement of momentum on all kinds of neural networks, but rather give a **simple setting where one can formally characterize the mechanism behind why momentum may improve generalization and for what type of datasets**. We believe that this point is well-enhanced in the introduction.
>
> In our real-world experiment, we are NOT trying to explain "why momentum works on CIFAR-10 data", rather we want to justify: If we have real-world data with some small margin data (for example photos taking in the rainy days), then its advised to use momentum to learn those small margin data better -- As shown in our theory.
>
>
>
>
> -- I am not convinced that taking a batch size of 1024 in SGD reduces it to GD.
>
> We have run additional experiments in the **full batch** setting for Figure 1 (resnet-18 on cifar10). What we observed is that GD+M gets 84.68% (against 85.20% when the batch size was 1024) and GD gets 75.83% (against 76.27%). The test accuracies are approximately similar and therefore, the conclusions are the same as the ones in the paper.
>
> -- It is unclear why the authors consider their setup more relevant to deep learning than the latter.
>
> We agree that the teacher-student setting with Gaussian input is also a great starting point. However, as shown in Table 1, momentum does not improve generalization in the case of fully connected and convolutional neural networks. Therefore we have to look for other settings beyond Gaussian input.
>
> Moreover, as we explained in the paper, in a image classification dataset such as CIFAR-10, we have patches that are informative (like the patch which contains the nose of a dog) and others that are not (like the patches containing the background). Besides, in some images in CIFAR-10, it is easy to recognize the category of these images thanks to these informative patches and for others, it is hard because the informative patch is blurry. Our dataset is a mathematical translation of this observation. Easy-to-recognize images are large margin data and hard-to-recognize images are small margin data. We believe that the fact that there are a lot of large margin data and fewer small margin data explains why momentum improves generalization.
>
> -- In the main body of the paper, I could only find experiments with ResNet and Wide-ResNet on CIFAR datasets, which does not seem like "many deep learning tasks".
>
> We apologize for the wording and thank the reviewer for noticing this. It's a typo we wanted to say "standard deep learning tasks" instead of "many deep learning tasks". We corrected this in the revised paper.
>
> -- The authors did not provide details on their experimental setup.
>
> We described it  in the captions of the figures (see Figure 1,2,4). We describe it in a more detailed way in a separate section in the revision of the paper.
>
> [1] Li, Y., et al. Algorithmic regularization in over-parameterized matrix sensing and neural networks with quadratic activations.
>
> [2] Woodworth, B., et al. Kernel and rich regimes in overparametrized models.
>
> [3] Allen-Zhu, Z., et al. Backward feature correction: How deep learning performs deep learning.

---

### Official Review · Reviewer_Vuuz · 2021-11-03

**Correctness:** 4
**Technical Novelty And Significance:** 2
**Empirical Novelty And Significance:** 2
**Recommendation:** 5
**Confidence:** 4

**Main Review:**

Strengths: The paper was clear and well written, particularly the toy example on when momentum hurts, as well as all of the proof sketches.

Weaknesses: My complaints are primarily around significance and presentation.

First some comparatively minor nits, to get them out of the way:
-The loss values in Figure 1 on the left pane are quite difficult to parse--you might consider making this logscale.
-In a couple of places you refer to "his momentum" when you probably mean "its momentum"

More substantively, I'm a little perplexed why the authors went to such great lengths to construct a dataset and prove statements about its training performance, but then did not actually perform any training using that dataset or that (peculiar) architecture used in the analysis of the theorems.  I think the manuscript would be greatly improved by explicitly demonstrating the generalization gap proved by the authors, preferably in a plot as a function of $d$ and  $\mu$ (which seem to be the key control knobs on the generalization results).

I'm likewise a bit unsure of the significance of this work.  It's not clear whether the mechanism proposed by the authors is also at play in other architectures, or if this is entirely a manifestation of the author's construction.  That is to say, the result is clearly progress, but it would significantly improve the impact if a thread could be drawn between this mechanism, and a non-synthetic dataset.

Finally, I'm a bit wary of ICLR submissions that have 60 pages of proofs.  These sorts of articles probably get more thorough review (and more _useful_ review) via a journal submission.


**Summary Of The Paper:**

The authors provide a new perspective on the why momentum is useful for generalization in neural networks.  They provide motivating intuition around why momentum is not always useful (including a great toy example), as well as empirical experiments on CIFAR-10.

Finally, the authors prove a variety of theorems on a synthetic problem where momentum provably results in greater generalization.

**Summary Of The Review:**

The work is technically sound (to my knowledge), but of limited significance.  This thread of work _could_ be of great significance if the mechanism proposed by the authors can be shown to be behind the performance of more realistic models, so I don't want to discourage the author's current line of investigation---it's more that I'm unsure if this is a good match for ICLR.

---

> ### Author Response · Authors · 2021-11-23
> **Response to Reviewer Vuuz**
>
> We thank the reviewer Vuuz for their review. We would like to clarify some points in the paper to ensure that the purpose of the paper is well-understood.
>
> **-- Experiments on the toy setting.**
>
> We agree that verifying the proposed simplified setting is numerically valid is a good idea. We thank the reviewer for this idea. We have run the experiment in the synthetic setting and added it in the revised paper (see Figure 3). We see that the GD+M model is able to better-classify the small margin data compared to GD, while on large margin data they behave identically.
>
> **-- peculiar neural networks.**
>
> We point out that neural networks with polynomial activation are standard in the deep learning theory community. A lot of prior theoretical works focus on neural networks with quadratic/cubic activations [1,2,3], and they are known to perform comparably with neural networks using ReLU activations [3].  Moreover, for a cubic activation function, $a(<w, x>)^3 = <a^{1/3}w  ,x >^3$. Therefore, training the second layer is not needed (can be absorbed in the hidden layer) so we simply set them to be one.
>
> **-- mechanism proposed by the authors is also at play in other architectures?**
>
> We point out that proving such a result as in our paper is extremely challenging for general deep learning models, and all the cited works in our paper regarding algorithmic regularization --which is the field of proving that some algorithms yield better generalization compared to others-- are **only studied in simplified setting of structured data and two-layer neural networks/linear learners**. We do empirically verify that the mechanism proposed in our paper works for real-life neural networks (as Resnets) in image classification --as detailed in the next point.
>
> **-- thread could be drawn between this mechanism, and a non-synthetic dataset?**
>
> The goal of the paper is to propose a theory for understanding the **mechanism** behind momentum improving generalization in image classification. The key mechanism of the simplified setting is the existence of small and large margin data that share the same feature and momentum helps learning the smaller margin ones. **We verified such mechanism for practical neural networks over a modified real-world CIFAR-10 data set with small/large margin data in Figure 4**, and show that having momentum indeed helps a lot in terms of learning small margin data.
>
> **--length of the submission.**
>
> Sure, we will reduce the size of the proofs and make them more compact in the last version.
>
>
> [1] Li, Y., et al. Algorithmic regularization in over-parameterized matrix sensing and neural networks with quadratic activations.  2018.
>
> [2] Woodworth, B., et al. Kernel and rich regimes in overparametrized models. 2020.
>
> [3] Allen-Zhu, Z., et al. Backward feature correction: How deep learning performs deep learning. 2020.

---

### Decision · Program_Chairs · 2022-01-20

**Decision:**

Reject

**Comment:**

All but one of the reviewers recommended rejecting this submission. The reviewer recommending acceptance (PBhC) was not confident in their assessment and was unwilling to champion the paper during the discussion phase, making it very difficult for me to unilaterally overrule the de facto reviewer consensus and recommend accepting the submission. Although some of the reviewers recommending rejecting the submission made relatively weak arguments, others raised more compelling points in favor of rejecting the paper. The discussion and reviews convinced me that the preponderance of the evidence indicated that I should recommend rejecting on the merits of the case anyway. Ultimately, I am recommending rejecting this submission, primarily because I do not believe the empirical contributions are strong enough, nor are they polished enough. Holistically, it is hard to see what impact this work can have without improved empirical evidence, given how little guidance the theoretical results give to practitioners. That said, I hope the authors iterate some more on the experiments and refocus the narrative a bit in that direction.

The paper exhibits a problem where gradient descent with momentum provably generalizes better than gradient descent without momentum. Given that momentum does not universally improve the out of sample error of neural networks trained with gradient descent, we should strongly suspect that there also exist problems where adding momentum to gradient descent degrades out of sample performance. Therefore, what actionable insights do we have? The paper suggests that perhaps the details of the problem (constructed in the submission) where momentum helps gives us an ability to predict when momentum will be helpful in practice, but we would need to see several more successful predictions of this form on typical datasets from the literature or other real (non-synthetic) datasets. Furthermore, has the literature and this submission even demonstrated convincingly enough that momentum improving out of sample error for the same training loss is a common occurrence? And has this submission even made a convincing empirical case on CIFAR10, let alone a larger selection of problems? The latter question would be sufficient to reject the submission, but resolving it favorably would not, in my view, be sufficient to accept the submission without also more evidence for the prevalence of this momentum generalization phenomenon or without demonstrating successful predictions about relative generalization performance on more problems.

Has the literature established that gradient descent or minibatch stochastic gradient descent often generalizes better when using momentum? The paper says "While these works shed light on how momentum acts on neural network training, they fail to capture the generalization improvement induced by momentum (Sutskever et al., 2013)." but Sutskever et al. to my recollection only measures training set loss and never properly considers questions of generalization. Certainly, in many places in the literature we see momentum get better validation error, but rarely do we get information on whether it does so for the same training loss and a priori we should suspect optimization speed is the primary effect at play. The paper also claims "Although it is well accepted that Momentum improve generalization in deep learning...", but the submission does not provide enough evidence that this is well accepted. The results of Leclerc & Madry (2020) are equivocal and may well be confounded by batch norm, but would need to be investigated further. So no, at least with the citations in this submission, it is far from well-established that momentum often improves generalization performance, i.e. that momentum results in better validation loss for the same training loss. Of course it won't always do this, but we should observe it regularly in the wild (the more dramatically the better) for this to be interesting.

Ok, but what about the experiments on CIFAR10? These experiments are hard to interpret because they seem to compare misclassification error (zero one loss) with the actual optimization objective of cross entropy error. These issues may be resolvable, but in their current form leave open too many loose ends. Just because two training runs both get zero classification errors on the training set does not mean that they do not differ in the log loss and even a small difference in log loss might explain a large difference in out of sample classification error. Although we often use these quantities as proxies for each other, that isn't quite safe and a better way to conduct this measurement would be to select an iterate of GD without momentum that has an almost identical (but slightly better) training cross entropy loss than a specific iterate of GD with momentum and then compare the cross entropy loss on the validation set, repeating for many different runs and iterates.

In the final analysis, stochastic gradient descent without momentum rarely gets used in practice and full gradient descent even more rarely, so this submission needs to do a better job of making a case for the impact it will have on researchers in this field. Perhaps a stronger case can be made, but I do not quite find the current version sufficiently compelling.